# Cross-platform comparison of arbitrary quantum states

D. Zhu[1,2,3,4,9], Z. P. Cian [1,2,5,9] ✉, C. Noel [1,2,5,6,7], A. Risinger[1,2,3], D. Biswas [1,2,5], L. Egan[1,2,5], Y. Zhu[1,5], A. M. Green [1,5], C. Huerta Alderete [1,5], N. H. Nguyen[1,5], Q. Wang[1,2,8], A. Maksymov[4], Y. Nam [4,5], M. Cetina[1,2,5,6], N. M. Linke [1,5], M. Hafezi [1,2,3,5] & C. Monroe[1,2,4,5,6,7]

As we approach the era of quantum advantage, when quantum computers (QCs) can outperform any classical computer on particular tasks, there remains the difficult challenge of how to validate their performance. While algorithmic success can be easily verified in some instances such as number factoring or oracular algorithms, these approaches only provide pass/fail information of executing specific tasks for a single QC. On the other hand, a comparison between different QCs preparing nominally the same arbitrary circuit provides an insight for generic validation: a quantum computation is only as valid as the agreement between the results produced on different QCs. Such an approach is also at the heart of evaluating metrological standards such as disparate atomic clocks. In this paper, we report a cross-platform QC comparison using randomized and correlated measurements that results in a wealth of information on the QC systems. We execute several quantum circuits on widely different physical QC platforms and analyze the cross-platform state fidelities.

Cross-platform quantum state comparisons are critical in the early stages of developing QC systems, as they may expose particular types of hardware-specific errors and also inform the fabrication of next-generation devices. There are straightforward methods for comparing generic output from different quantum computers, such as coherently swapping information between them[1–5], and full quantum state tomography[6]. However, these schemes require either establishing a coherent quantum channel between the systems[7], which may be impossible with highly disparate hardware types; or transforming quantum states to classical measurements, requiring resources that scale exponentially with system size.

Recently, a new type of cross-platform comparison based on randomized measurements has been proposed[8,9]. While this approach still scales exponentially with the number of qubits, it has a significantly

smaller exponent prefactor compared with full quantum state tomography[10], allowing scaling to larger quantum computer systems.

Here, we demonstrate a cross-platform comparison based on randomized-measurement[8,9,11], obtained independently over different times and locations on several disparate quantum computers built by different teams using different technologies, comparing the outcomes of four families of quantum circuits.

To quantify the comparison, we use the cross-platform fidelity defined as[8,12]

$$\mathcal{F}(\rho_1, \rho_2) = \frac{\text{tr}[\rho_1 \rho_2]}{\sqrt{\text{tr}[\rho_1^2]\text{tr}[\rho_2^2]}}, \tag{1}$$

[1]Joint Quantum Institute, University of Maryland, College Park, MD 20742, USA. [2]Center for Quantum Information and Computer Science, University of Maryland, College Park, MD 20742, USA. [3]Department of Electrical and Computer Engineering, University of Maryland, College Park, MD 20742, USA. [4]IonQ, College Park, MD 20740, USA. [5]Department of Physics, University of Maryland, College Park, MD 20742, USA. [6]Duke Quantum Center and Department of Physics, Duke University, Durham, NC 27708, USA. [7]Department of Electrical and Computer Engineering, Duke University, Durham, NC 27708, USA. [8]Chemical Physics Program and Institute for Physical Science and Technology, University of Maryland, College Park, MD 20742, USA. [9]These authors contributed equally: D. Zhu, Z. P. Cian. ✉e-mail: zpcian@umd.edu

where $\rho_i$ is the density matrix of the desired $N$ qubits quantum state produced by system $i$. To evaluate this fidelity, for each system, we first initialize $N$ qubits in the state $|0, 0, \ldots, 0\rangle$ and apply the unitary $V$ to nominally prepare the desired quantum states on each platform. In order to perform randomized-measurement, we measure the quantum states in $M_U$ different bases. In particular, we sample $M_U$ distinct combinations of random single-qubit rotations $U = u_1 \otimes u_2 \otimes \cdots \otimes u_N$ and append them to the circuit that implements $V$ as shown in Fig. 1a. Finally, we perform projective measurements in the computational basis. For each rotation setting $U$, the measurements are repeated $M_S$ times("shots") on each platform. We infer the cross-platform fidelity defined in Eq. (1) from the randomized measurement results via either the statistical correlations between the randomized measurements[8] (Protocol I in Method) or constructing an approximate classical representation of a quantum state using randomized measurements, the so-called the classical shadow[11,13] (Protocol II in Method).

We use four ion-trap platforms, the University of Maryland (UMD) EURIQA system[14] (referred to as UMD_1), the University of Maryland TIQC system[15] (UMD_2), and two IonQ quantum computers[16,17] (IonQ_1, IonQ_2), as well as five separate IBM superconducting quantum computing systems hosted in New York, *ibmq_belem* (IBM_1), *ibmq_casablanca* (IBM_2), *ibmq_melbourne* (IBM_3), *ibmq_quito* (IBM_4), and *ibmq_rome* (IBM_5)[18]. See Supplementary Information Sec. S4 for more details of these systems, which includes refs. 14, 18–22.

We first demonstrate the application of randomized measurements for comparing 5-qubit GHZ (Greenberger-Horne-Zeilinger) states[23] generated on different platforms and the ideal 5-qubit GHZ state obtained from classical simulation. Using the same protocol, we also compare states generated with three random circuits of different width and depth, each sharing a similar construction to circuits used in quantum volume (QV) measurements[24].

## Results

We first measure the cross-platform fidelity to compare 5-qubit GHZ states. Specifically, the circuit that prepares the GHZ states is appended with a total of $243 = 3^5$ different sets of single-qubit Clifford gates. These appended circuits complete all the measurements needed for quantum state tomography. Each appended circuit is repeated for $M_S = 2000$ shots. We sample $M_U = 100$ out of the 243 different $U$s to calculate the cross-platform fidelity defined in Eq. (1) (Fig. 1d). We see that our method has good enough resolution to reveal the performance difference between platforms. In Supplementary Information Sec. S2, we benchmark our method against full quantum state tomography by computing the fidelity as a function of $M_U$. The comparison shows that the fidelity obtained via randomized measurements approaches that obtained via the full quantum state tomography rapidly.

We present cross-platform fidelity results for 7- and 13-qubit QV-like circuits[24]. QV circuits have been studied extensively, both theoretically and experimentally[24–26], making them an ideal choice for the cross-platform comparison. Also, quantum volume provides a single-number metric for the overall performance of a quantum computer. However, in our randomized measurement scheme, we can obtain more information for the state we prepare. In particular, by using the classical post-processing scheme presented in[11], we can estimate many observables from the randomized measurement data. An $N$-qubit QV circuit consists of $d = N$ layers : each layer contains a random permutation of the qubit labels, followed by random two-qubit gates among every other neighboring pair of qubits. In our study, we call circuits of such construction but different circuit depth $d$ QV-like circuits. Specifically, a QV-like circuit can be written as a unitary operation $V = \prod_{i=1}^{d} V^{(i)}$, where $V^{(i)} = V_{\pi_i(N'-1),\pi_i(N')}^{i} \otimes \cdots \otimes V_{\pi_i(1),\pi_i(2)}^{i}$ and $N' = 2\lfloor N/2 \rfloor$. The operation $\pi(a)$ is a random permutation sampled from the permutation group $S_N$. The unitary operation $V_{a,b}^{i}$ is a random two-qubit gate acting on qubits $a$ and $b$ and sampled from $SU(4)$. The circuit diagram of an example QV-like circuit is shown in Fig. 2a. In this experiment, we infer the fidelity for 7-qubit QV-like states with $d = 2$ and $d = 3$ and a 13-qubit QV-like state with $d = 2$.

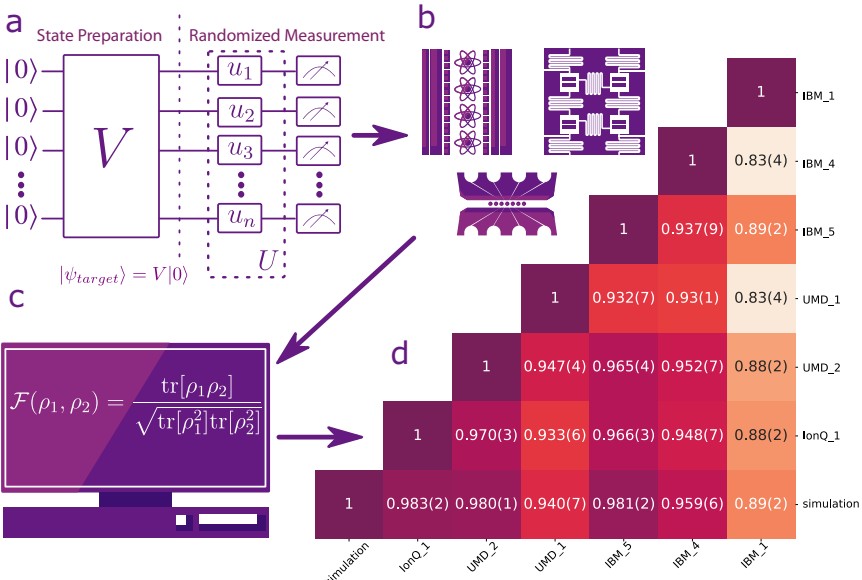

**Fig. 1 | Schematic diagram of the cross-platform comparison. a** Test quantum circuit, represented by unitary operator $V$ for state preparation, with appended random rotations $u_i$ to each qubit $i$ for measurements in a random (particular) basis. **b** The circuits are transpiled for different quantum platforms into their corresponding native gates. Each of the $M_U$ circuits is repeated $M_S$ times for each platform. **c** The measurement results are sent to a central data repository for processing the fidelities defined in Eq. (1). As an example, **d** The cross-platform fidelity results for a 5-qubit GHZ state, including a row of comparisons between each of the six hardware systems and theory (labeled "simulation"). Entry $i$, $j$ corresponds to the cross-platform fidelity between platform-$i$ and platform-$j$. The cross-platform fidelity is inferred from $M_U = 100$ randomized measurements and $M_S = 2000$ repetitions for each $U$.

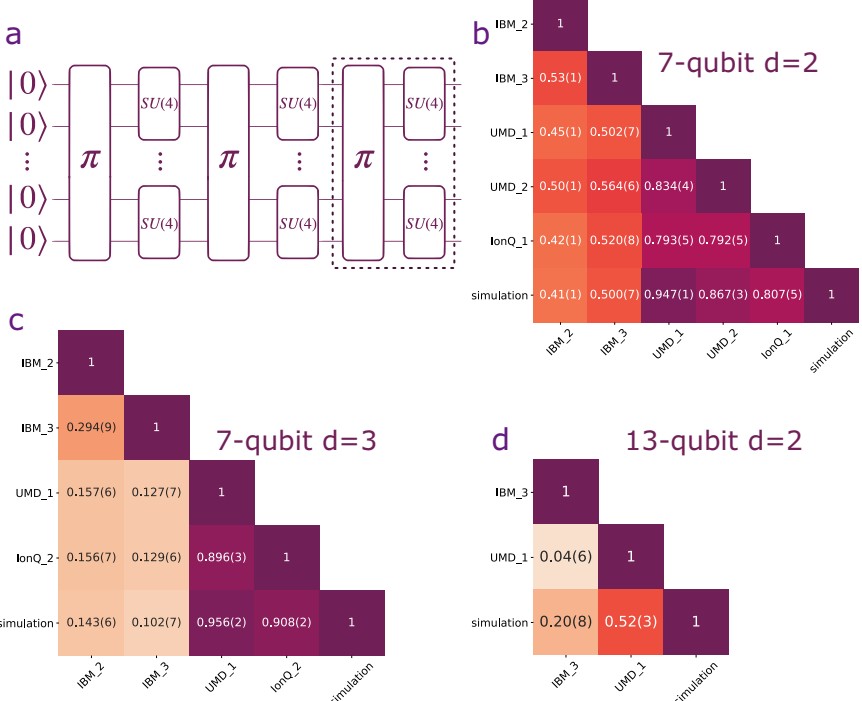

**Fig. 2 | The cross-platform fidelity for 7-qubit and 13-qubit QV-like circuit. a** The quantum volume circuit diagram for $d=3$. The $d=2$ case does not have the operations in the dashed rectangle. **b–d** Cross-platform fidelity between different quantum computers. Entry $i$, $j$ corresponds to the cross-platform fidelity $\mathcal{F}(\rho_i, \rho_j)$ between platform-$i$ and platform-$j$ as defined in Eq. (1). **b** $N=7$ and $d=2$; **c** $N=7$ and $d=3$; **d** $N=13$ and $d=2$.

Similar to the GHZ case, we first distribute the circuits, synthesize them into device-specific native gates, and allow optimizations/error-mitigation that satisfies the aforementioned state-preparation rule.

On each platform, we append the circuit with $M_U = 500$ different $U$s sampled using the greedy method. Outcomes are measured in the computational basis for $M_S = 2000$ shots. The cross-platform fidelities for $d=2$ and $d=3$ are shown in Fig. 2b, c. Our results verify that with only a fraction of the number of measurements required to perform full quantum state tomography, we can estimate the fidelities to sufficiently high precision to be able to see clear differences among them.

We also infer the cross-platform fidelity with a 13-qubit QV-like circuit with $d=2$. The results are shown in Fig. 2d. Here we use $M_U = 1000$ and $M_S = 2000$, in contrast with the much larger $M_U = 3^{13} = 1,594,323$ needed for full quantum state tomography.

We find several interesting features by analyzing the cross-platform fidelity of 7-qubit QV-like results. First, we observe that the cross-platform fidelity drops significantly when the number of layers $d$ increases from $d=2$ to $d=3$ for the IBM quantum computers. The drop may be due to the restricted nearest-neighbor connectivity of super-conducting quantum computers[27], requiring additional SWAP gates overhead for the execution of the permutation gates. In Supplementary Information Sec. S3, we numerically evaluate the number of entangling gates as function of the number of layers $d$ with different connectivity graphs. We see that, according to IBM's native compiler QISKit (see Supplementary Information Sec. S3 and Sec. S6 for measurement error calibration) extra entangling gates are used to perform two-qubit gates for non-nearest-neighbor qubits on superconducting platforms, resulting in extra errors.

The cross-platform fidelity between IBM_2 and IBM_3 is higher than the cross-platform fidelity between either of them and the ion-trap systems (and classical simulation) as shown in Fig. 2c. This motivates us to study whether quantum states generated from different devices tend to be similar to each other if the underlying technology of the two devices is the same. Therefore, we perform a further analysis to

investigate this phenomenon, which we refer to as intra-technology similarity.

We first study the fidelity between subsystems of the 7-qubit QV-like states prepared on different quantum computers for both $d=2$ and $d=3$. The subsystem fidelity provides a scalable way to estimate the upper bound for the full system fidelity, since the cost of measuring all possible subsystem fidelities of a fixed subsystem size scales polynomially with the full system size. For a given subsystem, we use the same data collected for the full system, but trace out qubits not within the subsystem of interest. The results are presented in Fig. 3a. We observe that the cross-platform fidelity between for all subsystem sizes from the same technology is higher for a given subsystem size.

To further characterize the intra-technology similarity, we perform principal component analysis[28] (PCA) on the randomized measurement data for the 7-qubit quantum volume states with $d=2$ and $d=3$ from all the platforms. PCA is commonly used to reduce the dimensionality of a dataset. It has been applied extensively in signal processing such as human face recognition and audio compression. When implementing PCA, we project the dataset onto the first few principal components to obtain lower-dimensional data while preserving as much of the variation as possible.

To prepare the data for PCA, we randomly sample 1000 shots from the randomized measurement data out of $M_U \times M_S = 1,000,000$ for each platform. We identify the set of Pauli strings whose expectation values can be evaluated using the sample. We then evaluate the expectation value of these identified Pauli strings by taking the average over the samples, and repeat the sampling $N_{sample} = 500$ times without replacement to make $N_{sample}$ data points in the $4^N$ dimensional feature space. The feature vectors represent averaged classical shadow of the quantum state generated from the quantum computers[11,29]. We perform a rotation on the feature space and find the first two principal axes, which are the axes that show the two most significant variances on the dataset. Figure 3b shows the projection of the $N_{sample}$ data points to the first two principal axes. We observe that the first principal

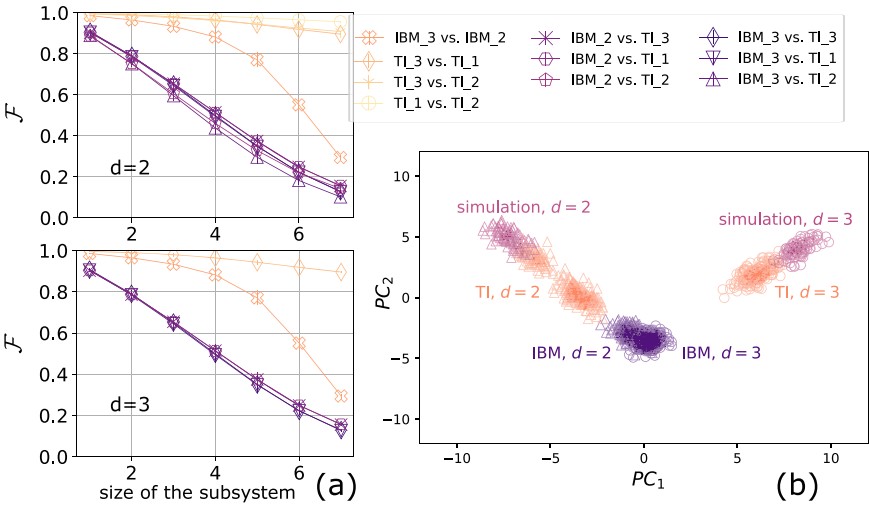

**Fig. 3 | The cross-platform fidelity for subsystem and intra-technology similarity. a** The cross-platform fidelity between subsystems prepared on different quantum computers. Left : 7-qubit quantum volume circuit of two layers. Right: 7-qubit quantum volume circuit of three layers. The mean for each subsystem size is calculated via bootstrap re-sampling. **b** The projection of randomized measurement dataset onto the first two principal axes, $PC_1$ and $PC_2$. Triangle marker is the 7-qubit quantum volume state with $d = 2$. Circle marker is the 7-qubit quantum volume state with $d = 3$. Magenta, orange, and violet correspond to simulation, trapped-ion, and IBM systems respectively.

component separates the two quantum volume states, and the second principal component can distinguish the technology that generates the states. The clustering of the data from the same technology indicates that each technology may share similar noise characteristics that can be distinguished through the cross-platform fidelity and machine-learning techniques.

## Discussion

In this manuscript, we experimentally performed the cross-platform comparison of four quantum states allowing the characterization of the quantum states generated from different quantum computers with significantly fewer measurements than those required by full quantum state tomography. To expand our understanding of the intra-technology similarity, more quantum states, in particular those designed to probe the effect of different settings on the cross-platform comparison results, should be studied. Our method could be extended to additional technological platforms such as Rydberg atoms and photonic quantum computers[30]. With the large volume of quantum data generated from the randomized measurement protocol, we have only begun to explore the possibilities that machine learning techniques can offer. We envision extensions of our method will be indispensable in quantitatively comparing near-term quantum computers, especially across different qubit technologies.

## Methods

### Inference of cross-platform fidelity

Here we briefly introduce the two protocols used for inferring cross-platform fidelity (Eq. 1) from randomized measurements. In Protocol I, we calculate the second-order cross-correlations[8] between the outcomes of the two platforms $i$ and $j$ via the relation

$$\text{Tr}[\rho_i \rho_j] = 2^N \sum_{s,s'} (-2)^{-D[s,s']} \overline{P_U^{(i)}(s) P_U^{(j)}(s')}, \quad (2)$$

where $i, j \in \{1, 2\}$, $s = s_1, s_2, \ldots, s_N$ is the bit string of the binary measurement outcomes $s_k$ of $k$th qubit, $D[s,s']$ is the Hamming distance between $s$ and $s'$, $P_U^{(i)}(s) = \text{Tr}[U\rho_i U^\dagger |s\rangle\langle s|]$, and the overline denotes the average over random unitaries $U$.

For Protocol II, we reconstruct the classical shadow of the quantum state for each shot of measurement as $\hat{\rho} = \bigotimes_{k=1}^N (3u_k^\dagger |s_k\rangle\langle s_k| u_k - I)$, where $I$ is the $2 \times 2$ identity matrix[11,13]. The overlap can

be calculated as[11]

$$\text{Tr}[\rho_i \rho_j] = \overline{\text{Tr}[\hat{\rho}_i \hat{\rho}_j]}, \quad (3)$$

where $i, j \in \{1, 2\}$ and the overline denotes the average over all the experimental realizations. We note that, for both protocols, unbiased estimators are necessary when calculating the purity $i = j$[8,11] using Eqs. (2) and (3).

While the fidelity inferred from the two protocols is identical in the asymptotic limit with $M = M_S \times M_U \to \infty$, the fidelity error inferred from Protocol II converges faster in the number of random unitaries[11]. Therefore, we implement Protocol II for 5- and 7-qubit experiments. However, this protocol is more costly for post-processing. Therefore, for the 13-qubit experiment, we post-process the result with Protocol I.

We explore two different schemes for sampling the single-qubit unitary rotations $U$, a random method and a greedy method. In the regime $M_S \gg 2^N$, we observe that the greedy method outperforms the random method (see Supplementary Information Sec. S1, which includes refs. 8, 11, 31). Therefore, for $N = 5, 7$, we sample the single-qubit unitary operation with the greedy method. For $N = 13$, we use the random method because to satisfy $M_S \gg 2^N$, the total number of measurements becomes too large. The specified target states and rotations are sent to each platform as shown in Fig. 1b, c. The circuit that implements the specified unitary $UV$ are synthesized and optimized for each platform in terms of its native gates.

When preparing a quantum state on a quantum system, one can perform various error-mitigation and circuit optimization techniques. While these techniques can greatly simplify the circuit and reduce the noise of the measurement outcomes, they can make the definition of state preparation ambiguous. For example, when we prepare a GHZ state and perform the projective measurement in the computational basis, we can defer the CNOT gates right before the measurement to the post-processing, instead of physically applying them. Although one can still obtain the same expectation value for any observable using such a circuit optimization technique, the GHZ state is not actually prepared in the quantum computer. In order to standardize the comparison, in this study, we require that one can perform arbitrary error-mitigation and circuit optimization techniques provided that the target state $|\psi_{target}\rangle = V|0\rangle$ is prepared at the end of the state-preparation stage.

After performing the experiments, the results are sent to a data repository. Finally, we process the results and calculate the cross-platform fidelities. The statistical uncertainty of the measured fidelity is inferred directly from the measurement results via a bootstrap resampling technique[32]. The bootstrap resampling allows us to evaluate the statistical fluctuation of the measurements together with the system performance fluctuation within the duration of the data taking, which is typically two to three days.

## Data availability
The data that support the findings of this study are available from the corresponding author upon request.

## Code availability
The code used for the analyses is available from the corresponding author upon request.

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

## Acknowledgements

We acknowledge Andreas Elben, Behtash Babadi, Benoi^t Vermersch and Peter Zoller for helpful discussions. We acknowledge the use of IBM Quantum services; the views expressed are those of the authors, and do not reflect the official policy or position of IBM or the IBM Quantum team. This work was supported by the ARO through the IARPA LogiQ program (11IARPA1008), the NSF STAQ Program (PHY-1818914), the AFOSR MURIs on Dissipation Engineering in Open Quantum Systems (FA9550-19-1-0399) and Quantum Interactive Protocols for Quantum Computation (FA9550-18-1-0161), the ARO MURI on Modular Quantum Circuits (W911NF1610349), and the U.S. Department of Energy Quantum Systems Accelerator (QSA) Research Center (DE-FOA-0002253) and National Science Foundation QLCI grant OMA-2120757. N.M.L. acknowledges support from the Maryland-Army-Research-Lab Quantum Partnership (W911NF1920181), the Office of Naval Research (N00014-20-1-2695), and the NSF Physics Frontier Center at JQI (PHY-1430094). A.M.G. is supported by a JQI Postdoctoral Fellowship.

## Author contributions

D.Z., Z.P.C., C.N., Y.N., M.C., N.M.L., M.H., and C.M. designed the research; D.Z., Z.P.C., C.N., A.R., D.B., L.E., Y.Z., A.M.G., C.H.A., N.H.N., Q.W., and A.M. performed experiments and collected the data; Z.P.C., A.M., and Y.N. compiled and optimized the circuit; D.Z. and Z.P.C. analyzed data; D.Z., Z.P.C., C.N., Q.W., Y.N. N.M.L., M.H., and C.M. contributed to the manuscript, with input from all authors.

## Competing interests

C.M., A.M., and Y.N. are affiliated with IonQ. All other authors declare that they have no competing interests.
