## [Peer Review File · Nature Communications]

REVIEWER COMMENTS

Reviewer #1 (Remarks to the Author):

Overall, the authors have sufficiently addressed the questions I raised in my previous review. Hence, I would recommend acceptance. In the following, I will present my previous review along with some comments (indicated by >>) based on the feedback from the authors.

The paper provides a study that benchmarks the relation between different quantum computing systems. The proposed method offers a similarity graph between different quantum computing systems. The manuscript compares four ion trap systems and five superconducting qubit systems, a good variety over existing platforms. The benchmarking protocol is based on randomized measurements and can be seen as a synthesis of the method proposed in [1] and [2]. The obtained similarity measure between different computing systems is fed into an unsupervised machine learning approach (principal component analysis) to generate a low-dimensional representation for the considered quantum computing systems.

The cross-platform verification based on randomized measurements is nicely executed in this paper (an earlier proposal is given in [1]). And I like the unsupervised ML method for visualizing the relations between different quantum computing systems.

The method used to perform the experiments is currently the most dependable protocol to the best of my knowledge. A greedy method for choosing the unitary is proposed in this work that yields a slight boost in performance. I would suggest the authors give a more extensive discussion on the proposed greedy method. For example, the proposed greedy method resembles the derandomization method proposed in [3], which has been proven to always yield better (or at least equal) performance to the purely randomized method. Some further exploration or discussion on that front would make the work stronger.

>> The authors have included a new set of experiments comparing greedy methods with some other approaches. The authors show that the greedy method is indeed better in some regimes.

Using the randomized measurement data with unsupervised machine learning to give a clear visualization for the different quantum computing systems is an intriguing idea. The use of unsupervised machine learning over randomized measurements has been considered in [2]. But in [2], the purpose is to classify topological quantum phases, which is very different from the usage in this work. I am curious about some details of this approach. Does the choice of states change the representation significantly? For example, what if we consider the GHZ states to conduct PCA? Could we provide some interpretations for the two principal components? Including this information will be beneficial.

>> I agree with the statements presented by the authors. It does seem true that for different states, the intra-technology similarity is still useful for understanding different platforms.

Overall, I think the work demonstrates a nice set of experiments for benchmarking the similarity between different quantum computing systems. But I would like to see a stronger case made for using cross-platform comparison. I think the advantage of using cross-platform comparison and why we care about cross-platform comparison are not fully fleshed out in the current manuscript. The following presents some of my thoughts, challenges, and questions for using cross-platform benchmarking and, more specifically, those based on randomized measurements.

One advantage of cross-platform benchmarking is that one can probe the performance of quantum computing systems when we can no longer simulate a large number of qubits on a classical computer. If the above is the primary purpose for cross-platform benchmarking, here are some challenges. (1) What if all of our currently available technology has a similar source of errors, such that they seem to yield good cross-platform performance, but the quantum computing system is erroneous? (2) There are some trivial choices for performing large-scale benchmarking that don't require classical simulation of the quantum device. What if we implement a sizeable random circuit U , followed by running U^\dagger , then check if all the outputs are zero? What additional information would cross-platform benchmarking give us that the suggested trivial protocol could not?

>> I agree with the authors that the information obtained from running the proposed protocol would generally be distinct from the mirroring approach I suggested. Hence, it seems likely that the proposed approach could still be useful in practice.

The following are some "challenge" questions for using randomized measurements to do cross-platform benchmarking. (1) The strength of doing randomized local measurement utilized in this work is that it can adequately estimate the properties of the reduced density matrices. If the generated state is highly entangled (e.g., when depth d in the experiments is large), every reduced density matrix will look completely mixed. How should we perform cross-platform benchmarking in that case? (2) The fidelity estimates will scale exponentially with system size (albeit a small exponential factor). What should one do when the quantum computing system has a hundred qubits (which many seem to be optimistic in creating in the next few years)?

>> Regarding the questions I raised, there is a fundamental limit when one does not allow the two quantum states on two different platforms to interact coherently through a quantum network [5]. So I would say that this is not a particular problem with the proposed approach, but any approach that does not utilize a quantum network.

I believe addressing some of the above questions or offering new perspectives on the usefulness of cross-platform comparison will make the paper stronger.

>> I think the authors have provided sufficient results addressing my questions.

[1] Elben, Andreas, et al. "Cross-platform verification of intermediate scale quantum devices." *Physical review letters* 124, 010504 (2020).

[2] Huang, Hsin-Yuan, Richard Kueng, and John Preskill. "Predicting many properties of a quantum system from very few measurements." *Nature Physics* 16.10 (2020): 1050-1057.

[3] Huang, Hsin-Yuan, Richard Kueng, and John Preskill. "Efficient estimation of Pauli observables by derandomization." *Physical review letters* 127, 030503 (2021).

[4] Huang, Hsin-Yuan, et al. "Provably efficient machine learning for quantum many-body problems." *arXiv preprint arXiv:2106.12627* (2021).

[5] Anshu, Anurag, Zeph Landau, and Yunchao Liu. "Distributed quantum inner product estimation." *arXiv preprint arXiv:2111.03273* (2021).

Reviewer #2 (Remarks to the Author):

This paper demonstrates an experimental proof-of-principle study of two previously proposed protocols to measure cross-platform validation. Both protocols are based on randomized measurements. The authors use both methods to compare various trapped ion and superconducting qubit systems. Two different algorithms are tested on the different platforms: preparation of a GHZ state and a random circuit based on quantum volume measurements. The authors provide analysis of their results and use a machine learning approach to further analyze the data.

The main motivation of the paper seems to be that the two protocols are more efficient despite scaling exponentially as well. However, the fidelity of GHZ states can be estimated efficiently without the exponential scaling. What is the benefit of the methodologies used here over direct GHZ fidelity measurements?

Further, quantum volume itself was developed to have a direct comparison between different devices/platforms. On one hand, quantum volume requires pre-computing the ideal probability distribution, and while requiring exponential classical resources, ~30 qubits are still feasible. On the other hand, the methodologies presented here were only implemented on max 13 qubits. What is the benefit of these methods compared to just measuring quantum volume and comparing the outcomes of different platforms?

It would have been interesting if the authors could compare the two protocols to the above-mentioned comparisons (direct GHZ fidelity measurements and measuring quantum volume).

Generally, the paper is clearly written, and the implementation of the protocols and data acquisition seems valid. If the authors can make a stronger case for using the implemented protocols, I could support a publication here.

Reviewer #3 (Remarks to the Author):

The author's answers address my concerns, regarding usefulness of their methods compared to full quantum tomography for small to medium sized systems. Even though they are not scalable, the results prove the potential draw useful conclusions on different technologies and intra-technology similarity. Also glad the authors acknowledge the potential of extending their method.

Given that I am satisfied with the answers and having read the communication with the other referees, I believe the work is both significant and rigorous enough to be worthy of publication in Nature Communications.

Reviewer #1 (Remarks to the Author):

Overall, the authors have sufficiently addressed the questions I raised in my previous review. Hence, I would recommend acceptance. In the following, I will present my previous review along with some comments (indicated by >>) based on the feedback from the authors.

The paper provides a study that benchmarks the relation between different quantum computing systems. The proposed method offers a similarity graph between different quantum computing systems. The manuscript compares four ion trap systems and five superconducting qubit systems, a good variety over existing platforms. The benchmarking protocol is based on randomized measurements and can be seen as a synthesis of the method proposed in [1] and [2]. The obtained similarity measure between different computing systems is fed into an unsupervised machine learning approach (principal component analysis) to generate a low-dimensional representation for the considered quantum computing systems.

The cross-platform verification based on randomized measurements is nicely executed in this paper (an earlier proposal is given in [1]). And I like the unsupervised ML method for visualizing the relations between different quantum computing systems.

The method used to perform the experiments is currently the most dependable protocol to the best of my knowledge. A greedy method for choosing the unitary is proposed in this work that yields a slight boost in performance. I would suggest the authors give a more extensive discussion on the proposed greedy method. For example, the proposed greedy method resembles the derandomization method proposed in [3], which has been proven to always yield better (or at least equal) performance to the purely randomized method. Some further exploration or discussion on that front would make the work stronger.

>> The authors have included a new set of experiments comparing greedy methods with some other approaches. The authors show that the greedy method is indeed better in some regimes.

Using the randomized measurement data with unsupervised machine learning to give a clear visualization for the different quantum computing systems is an intriguing idea. The use of unsupervised machine learning over randomized measurements has been considered in [2]. But in [2], the purpose is to classify topological quantum phases, which is very different from the usage in this work. I am curious about some details of this approach. Does the choice of states change the representation significantly? For example, what if we consider the GHZ states to conduct PCA? Could we provide some interpretations for the two principal components? Including this information will be beneficial.

>> I agree with the statements presented by the authors. It does seem true that for different states, the intra-technology similarity is still useful for understanding different platforms.

Overall, I think the work demonstrates a nice set of experiments for benchmarking the similarity between different quantum computing systems. But I would like to see a stronger case made for using cross-platform comparison. I think the advantage of using cross-platform comparison and why we care about cross-platform comparison are not fully fleshed out in the current manuscript. The following presents some of my thoughts, challenges, and questions for using cross-platform benchmarking and, more specifically, those based on randomized measurements.

One advantage of cross-platform benchmarking is that one can probe the performance of quantum computing systems when we can no longer simulate a large number of qubits on a classical computer. If the above is the primary purpose for cross-platform benchmarking, here are some challenges. (1) What if all of our currently available technology has a similar source of errors, such that they seem to yield good cross-platform performance, but the quantum computing system is erroneous? (2) There are some trivial choices for performing large-scale benchmarking that don't require classical simulation of the quantum device. What if we implement a sizeable random circuit U , followed by running U^\dagger , then check if all the outputs are zero? What additional information would cross-platform benchmarking give us that the suggested trivial protocol could not?

>> I agree with the authors that the information obtained from running the proposed protocol would generally be distinct from the mirroring approach I suggested. Hence, it seems likely that the proposed approach could still be useful in practice.

The following are some "challenge" questions for using randomized measurements to do cross-platform benchmarking. (1) The strength of doing randomized local measurement utilized in this work is that it can adequately estimate the properties of the reduced density matrices. If the generated state is highly entangled (e.g., when depth d in the experiments is large), every reduced density matrix will look completely mixed. How should we perform cross-platform benchmarking in that case? (2) The fidelity estimates will scale exponentially with system size (albeit a small exponential factor). What should one do when the quantum computing system has a hundred qubits (which many seem to be optimistic in creating in the next few years)?

>> Regarding the questions I raised, there is a fundamental limit when one does not allow the two quantum states on two different platforms to interact coherently through a quantum network [5]. So I would say that this is not a particular problem with the proposed approach, but any approach that does not utilize a quantum network.

I believe addressing some of the above questions or offering new perspectives on the usefulness of cross-platform comparison will make the paper stronger.

>> I think the authors have provided sufficient results addressing my questions.

[1] Elben, Andreas, et al. "Cross-platform verification of intermediate scale quantum devices." Physical review letters 124, 010504 (2020).

- [2] Huang, Hsin-Yuan, Richard Kueng, and John Preskill. "Predicting many properties of a quantum system from very few measurements." *Nature Physics* 16.10 (2020): 1050-1057.
- [3] Huang, Hsin-Yuan, Richard Kueng, and John Preskill. "Efficient estimation of Pauli observables by derandomization." *Physical review letters* 127, 030503 (2021).
- [4] Huang, Hsin-Yuan, et al. "Provably efficient machine learning for quantum many-body problems." *arXiv preprint arXiv:2106.12627* (2021).
- [5] Anshu, Anurag, Zeph Landau, and Yunchao Liu. "Distributed quantum inner product estimation." *arXiv preprint arXiv:2111.03273* (2021).

We thank the reviewer for the insightful summary and the recommendations.

Reviewer #2 (Remarks to the Author):

This paper demonstrates an experimental proof-of-principle study of two previously proposed protocols to measure cross-platform validation. Both protocols are based on randomized measurements. The authors use both methods to compare various trapped ion and superconducting qubit systems. Two different algorithms are tested on the different platforms: preparation of a GHZ state and a random circuit based on quantum volume measurements. The authors provide analysis of their results and use a machine learning approach to further analyze the data.

The main motivation of the paper seems to be that the two protocols are more efficient despite scaling exponentially as well. However, the fidelity of GHZ states can be estimated efficiently without the exponential scaling. What is the benefit of the methodologies used here over direct GHZ fidelity measurements?

Further, quantum volume itself was developed to have a direct comparison between different devices/platforms. On one hand, quantum volume requires pre-computing the ideal probability distribution, and while requiring exponential classical resources, ~30 qubits are still feasible. On the other hand, the methodologies presented here were only implemented on max 13 qubits. What is the benefit of these methods compared to just measuring quantum volume and comparing the outcomes of different platforms? It would have been interesting if the authors could compare the two protocols to the above-mentioned comparisons (direct GHZ fidelity measurements and measuring quantum volume).

We agree that GHZ fidelity (the overlap between wave function generated by quantum computer and ideal GHZ state) can be estimated in polynomial time. However, the cross-platform fidelity between the wave function generated by two disparate quantum computers is generally more costly and requires exponential scaling with system size [10]. Here, we use GHZ states for one of our demos so that our method can be compared against different well studied efficient GHZ state characterization methods. So we could know what to expect for general non-GHZ states.

The benefit of this experiment is that in the absence of a quantum communication channel, thus forbidding the implementation of the SWAP test, one can still estimate the cross-platform fidelity between states prepared from distinct quantum computers. The resource for the estimation is

more efficient than that for the full state tomography. Also, quantum volume provides a single-number metric for the overall performance of a quantum computer. However, in our randomized measurement scheme, we can obtain more information for the state we prepare. In particular, by using the classical post-processing scheme presented in [11], we can estimate many observables from the randomized measurement data.

We include the sentence above “Also, quantum volume provides a single-number.....we can estimate more observables from the randomized measurement....” into page 8 of our manuscript, where the concept of quantum volume is first introduced.

First, in contrast to the quantum volume measurement, our scheme does not require classical simulation of a quantum circuit. Moreover, since the sample complexity of the cross-platform comparison is $O(\sqrt{d})$, making the task experimentally feasible for around 30 qubits. The number of qubits used in our experiment was only limited by the quantum computers we had access to.

Generally, the paper is clearly written, and the implementation of the protocols and data acquisition seems valid. If the authors can make a stronger case for using the implemented protocols, I could support a publication here.

We thank the reviewers for their insightful comments.

Reviewer #3 (Remarks to the Author):

The author's answers address my concerns, regarding usefulness of their methods compared to full quantum tomography for small to medium sized systems. Even though they are not scalable, the results prove the potential draw useful conclusions on different technologies and intra-technology similarity. Also glad the authors acknowledge the potential of extending their method.

Given that I am satisfied with the answers and having read the communication with the other referees, I believe the work is both significant and rigorous enough to be worthy of publication in Nature Communications.

We are grateful to the reviewer's helpful comment and the recommendations for publication.